# What if publication bias is the rule and net carbon loss from priming the exception?

Jennifer Michel[1,2,*], Yves Brostaux[3], Bernard Longdoz[2,4], Hervé Vanderschuren[1,2,5], Pierre Delaplace[1,2]

[1]Plant Genetics and Rhizosphere Processes laboratory, Gembloux Agro-Bio Tech, University of Liège, Gembloux, Belgium
[2]Transdisciplinary Agroecosystem Platform for Integrated Research (TAPIR), TERRA teaching and research centre, Gembloux Agro- Bio Tech, University of Liège, Gembloux, Belgium
[3]Modélisation et développement, Gembloux Agro-Bio Tech, University of Liège, Gembloux, Belgium
[4]Biosystems Dynamics and Exchanges (BIODYNE), TERRA teaching and research centre, Gembloux Agro-Bio Tech, University of Liège, Gembloux, Belgium
[5]Tropical Crop Improvement Lab, Department of Biosystems, KU Leuven, Heverlee, Belgium
*Correspondence to: Jennifer Michel (jennifer.michel@uliege.be)

**Abstract**

Priming effects in soil science describe the influence of fresh carbon (C) inputs on rates of microbial mineralisation of native soil organic matter, which can either increase (positive priming) or decrease (negative priming). While both positive and negative priming effects occur in natural ecosystems, the latter is less documented in the peer-reviewed literature and the overall impact of priming effects on the C balance of vegetated ecosystems remains elusive. Here, we highlight three aspects which need to be discussed to ensure (rhizosphere) priming effects are correctly perceived in their ecological context and measured at appropriate scales: (i) We emphasize the importance of evaluating net C balances because usually experimental C inputs exceed C-losses meaning even positive priming doesn't cause net C-loss; (ii) We caution against publication bias, which forces overrepresentation of positive priming effects, neglects negative or no priming, and potentially misguides conclusions about C-loss; and (iii) We highlight the need to distinguish between general priming effects and rhizosphere-specific priming, which differ in their scale and driving factors, and hence require different methodological approaches. Future research should focus on scalable experiments linking priming to plant nutrition via C, nutrient and water cycling to understand priming in context of ecosystem functioning.

## More nuance and context in (rhizosphere) priming papers is needed

Priming effects refer to changes in soil C mineralisation rates caused by exogenous C inputs to soil such as litter, while rhizosphere priming effects more specifically refer to the changes in soil microbial activity and nutrient cycling caused by root C inputs to soil by plants. The carbon (C) compounds in root exudates and litter can either stimulate microbial growth and metabolism, leading to increased mineralization of soil organic matter (positive priming), or decrease microbial soil mineralisation when microbes assimilate primarily plant-derived C (negative priming) (Kuzyakov et al., 2000; Blagodatskaya et al., 2011; Dijkstra et al., 2013). Both positive and negative priming effects are commonly reported in the literature, and they are not mutually exclusive in ecosystems (Bastida et

al. 2019; Feng & Zhu, 2021; Michel et al. 2024). In many studies, observations include both positive

and negative priming either depending on experimental condition, or sometimes substrate amendments also result in mixed positive, negative and/or no priming within one unique modality (Chen at al. 2014; Qiao at al. 2016; Heitkötter at al. 2017; Hicks at al. 2019; Michel et al., 2022). Individual priming effects are mostly short-term phenomena, but continuously occur in the rhizosphere of living plants, where active root exudation provides energy-rich labile C to soil

microbes, while rhizodeposition also supplies more complex substances like cellulose to the soil (Canarini et al. 2019; Villarino et al. 2021). While it is increasingly recognised that priming effects are an important mechanism to regulate plant nutrition, the impact of priming effects on the overall C balance remains controversial (Dijkstra et al., 2013; Zhu et al. 2014; Holz et al., 2023, Pausch et al., 2024). C inputs to soil can also directly interact with the abiotic soil matrix and C inputs can also

enter the soil food web bypassing microbes, which can cause temporal and spatial shifts in the priming response to fresh inputs (Barreto et al., 2024; Sokol et al., 2024; van Bommel et al. 2024). Here, we highlight three aspects which need to be discussed to ensure (rhizosphere) priming effects are correctly perceived in their ecological context and measured at appropriate scales to avoid a one-sided narrative distorted towards C loss caused by positive priming.

**(i)** The first aspect is that there is little empirical evidence for net C losses from priming as in most studies, including those reporting exclusively positive priming effects, the experimentally added quantities of C to the study system exceed the amounts lost in basal and primed respiration.

**(ii)** The second aspect is that publication bias is critical, with studies tending to overrepresent positive priming and inferring C loss without empirical evidence.

**(iii)** The third aspect is a lack of distinction between priming effects (PE) and rhizosphere priming effects (RPE) which are measured at different scales, have different drivers and therefore differ in their ecological interpretability.

**i) Even positive priming effects seldom cause net carbon loss**

Many studies focus on C losses from (positive) priming effects, which has been the historic narrative

in priming literature (e.g. Löhnis, 1926; Jenkinson et al. 1985). Positive priming and net C losses are observed in studies, but the number of studies with true C loss is relatively small as commonly the inputs exceed the outputs (Liang et al., 2018). Yet, the small number of studies reporting net C loss and stating huge implications for ecosystem C cycling has a disproportionally strong impact on the overall perception of priming because the results are "catchy", which can have a strong imprint on the

mind (Table 1). Nonetheless, recently more studies provided a more comprehensive view on C budgets and revealed that there is little evidence for net C loss from priming effects (Qiao et al., 2014; Liang et al., 2018; Siles et al., 2022; Qin et al., 2024; Chen et al. 2025). For example, a recent meta-analysis evaluating the impact of priming effects derived from crop residues and their interaction with

nitrogen inputs concluded that there was no C loss despite the positive priming reported (Qin et al.

2024; Figure 1). This finding aligns with assessments in many soil incubation studies which demonstrate a net C balance in favour of C sequestration because in these experiments the C inputs from labile substrates usually exceed the C outputs from basal and primed respiration by at least one order of magnitude (Qiao et al., 2014; Cardinael et al., 2015; Liang et al., 2018; Schiedung et al., 2023; Qin et al., 2024). In accordance with these observations in lab incubations, several studies

upscaling priming effects over longer time scales and to areas of several hectares also indicate that priming effects may not change overall C budgets. For example, Schiedung et al. (2023) evaluated priming effects along a 20-year chronosequence of land inversion in New Zealand to identify the dependence of priming effects on root-derived C in topsoil and sub soils. Even though positive priming was reported, overall, C losses with priming never exceeded new root-derived C inputs.

Similar observations were made by Yin et al. (2019) who studied rhizosphere priming effects and microbial biomass C dynamics of two wheat genotypes grown under two temperatures and found no net soil organic C loss or gain as C loss caused by higher RPE was counteracted by increased microbial growth/turnover. Similarly, Cardinael et al. (2015) used a 52-year long field experiment where SOC stocks of fallow fields were compared to SOC stocks of fields regularly receiving fresh or

composted straw to show that no significant difference in SOC stocks dynamics occurred over the 52 years, suggesting no long-term impact of priming effect. Equalising priming with C loss is hence not a valid conclusion and studies should consistently evaluate and present the experimental C inputs and outputs and report the net C balance to avoid misleading the reader to believe that priming imperatively has a strong impact on soil C budgets.

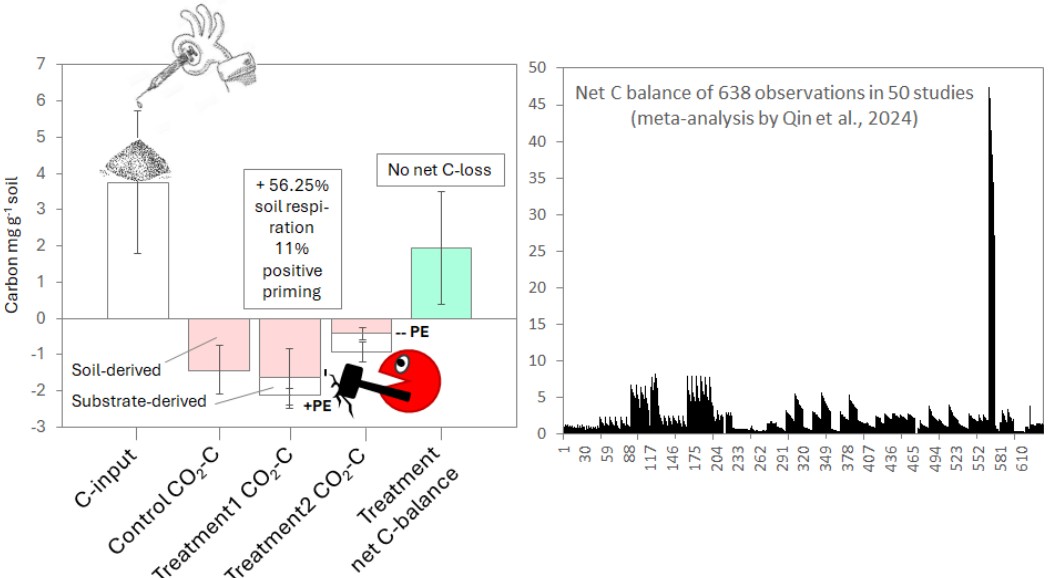

**Figure 1. Net C balance.** Left: Principle of C balance calculation (sum of C inputs minus sum of C output) on a common soil incubation data set with positive (treatment 1) and negative (treatment 2) priming, and no net C loss in neither case because a lot of added C input is not respired and hence stayed in the system either in microbial biomass or dissolved organic C. Right: Net C balance of the n=638 observations of n=50 priming studies included in the meta-analysis of Qin et al. (2024).

**Table 1: Cognitive and systemic biases which can influence perception of priming effects** (partly after Ruhl, 2023). For an objective analysis free of biases, the essential step is to be aware of the biases (by reading below table e.g.) and engage in discussion of a broader perspective.

| Cognitive and systematic biases | Definition | Example | Further reading |
|---|---|---|---|
| **Availability heuristic or availability bias** | Rare but vivid or emotionally striking cases disproportionately influence perceptions and narratives, overshadowing more common but less dramatic outcomes; "top of mind" thinking where the first information which comes to mind is taken as a general rule | "I read about HUGE carbon loss from priming in a paper in (insert big journal name) by (insert big scientist name) from (insert big institute name) and it is cited 10000000 times, it must be the general rule and super important." | Tversky & Kahneman, 1973 |
| **Confirmation bias** | Tendency to interpret new information as confirmation of preexisting beliefs and opinions while giving disproportionately less consideration to alternative possibilities; selectively read or remember information that supports preexisting beliefs and failure to seek out sources that challenge them; choose to reinforce preexisting ideas because being right helps preserve a sense of self-esteem, which is important for feeling secure in the world and maintaining positive relationships | "I have always thought that priming causes carbon loss and is a problem for the planet, of cause these results also show that." | Wason, 1960; Nickerson, 1998; Oswald & Grosjean, 2004 |
| **Hindsight bias** | Tendency to perceive past events as more predictable than they actually were; why we ascribe larger certainty to knowing the outcome of an event only once the event is completed | "I knew that would happen" | Jeng 2006; Roese & Vohs, 2012 |
| **Inattentional blindness** | Failure to notice factors outside the main focus | "I am focussed on priming effects and fail to look at the net carbon balance / unmetabolized inputs" | Most et al., 2001 |
| **Peer pressure** | Influence exerted by a social environment (peer group) to conform to the beliefs, behaviours, or expectations of the majority or the dominant voices; can result in suppression of dissenting opinions and group norms in conflict with available evidence | "All my colleagues exclusively publish positive priming, and in good journals, and they want to submit a proposal about it, I can impossibly report something else" | Asch, 1951; Cialdini & Goldstein, 2004 |

**ii) Cognitive and systemic biases cause overrepresentation of positive priming in the literature**

The dominance of positive priming in the literature may be inflated by cognitive and systemic biases, which can skew perceptions, research practices, and publication outcomes (Table 1). These biases, including availability heuristic, confirmation bias, hindsight bias, inattentional blindness, and peer pressure, systematically distort the scientific narrative, overemphasizing positive priming while underrepresenting neutral or negative effects. Understanding these biases is critical to foster a

balanced scientific discourse and accurately assess the global direction of priming effects. The availability heuristic leads researchers and readers to overestimate the prevalence of positive priming effects due to previous catchy or highly cited studies. For example, a widely publicised study in a prestigious journal claiming dramatic C loss from priming can become "top of mind," overshadowing more common studies showing minimal or no effects. This bias is compounded by confirmation bias,

where researchers may selectively interpret data to align with the prevailing narrative that priming causes significant C loss. For instance, a scientist who believes priming is a major environmental issue might focus on results supporting this view while dismissing contradictory evidence, reinforcing preconceived notions. Hindsight bias further distorts perceptions by making positive priming effects seem more predictable after they are reported. Researchers may claim they knew priming would lead

to C loss, even when earlier evidence was ambiguous, solidifying the narrative of positive priming as inevitable. Inattentional blindness contributes by causing researchers to overlook critical factors, such as net C balance or unmetabolized inputs, when focusing narrowly on priming effects. This tunnel vision can lead to incomplete interpretation of data, emphasizing certain outcomes while ignoring broader ecosystem dynamics. Peer pressure plays a significant role in perpetuating such biases, as

researchers face social and professional incentives to conform to dominant trends. This systemic pressure contributes to publication bias, where studies reporting positive priming are more likely to be submitted and accepted, while those showing neutral or negative effects are underrepresented, creating an asymmetrical body of literature. In meta-analysis, graphical tools like funnel plots are commonly used to detect publication bias (Figure 2). These plots display effect sizes (e.g. response

ratios) against a measure of study precision (e.g. standard error). Symmetrical plots suggest balanced reporting, while asymmetry - often with a skew toward positive effects - indicates potential bias, where smaller studies with large positive effects are overrepresented. High heterogeneity (e.g. $I^2 > 75\%$) in these analyses often reflects variability in study methods or selective reporting (aka biases), further complicating the synthesis of global priming effects. Corrective methods in meta-analysis such

as trim-and-fill can estimate missing studies to adjust effect sizes (Jennions & Møller, 2002). Applying such analysis to the data of a meta-meta-analysis on priming effects (by Xu et al., 2024) for example revealed an overall moderate priming estimate of 10.7% (estimated effect size (log-transformed response ratio) of 0.1022 (CI95: 0.0740, 0.1305)) rather than inflated figures like 125%). Given that none of the underlying meta-analysis has been corrected for publication bias, the actual

priming estimate be even lower. This analysis demonstrating that the interplay of cognitive and

systemic biases in scientific literature can strongly distort the representation of priming: When

availability heuristic and confirmation bias amplify attention to positive priming, hindsight bias

reinforces its perceived inevitability, inattentional blindness narrows focus to supportive data, and

peer pressure and publication bias suppress contradictory findings, this can lead to an exaggerated

narrative of C loss, potentially misinforming environmental policy and management. To address this,

researchers must prioritize transparency, encourage publication of neutral or negative results, and

critically evaluate methodological variability (Figure 3). By mitigating these biases, the scientific

community can develop a more accurate and balanced understanding of priming effects and their

implications not only for the global C cycle, but also for plant nutrient uptake and the regulation of

biogeochemical cycles in natural ecosystems.

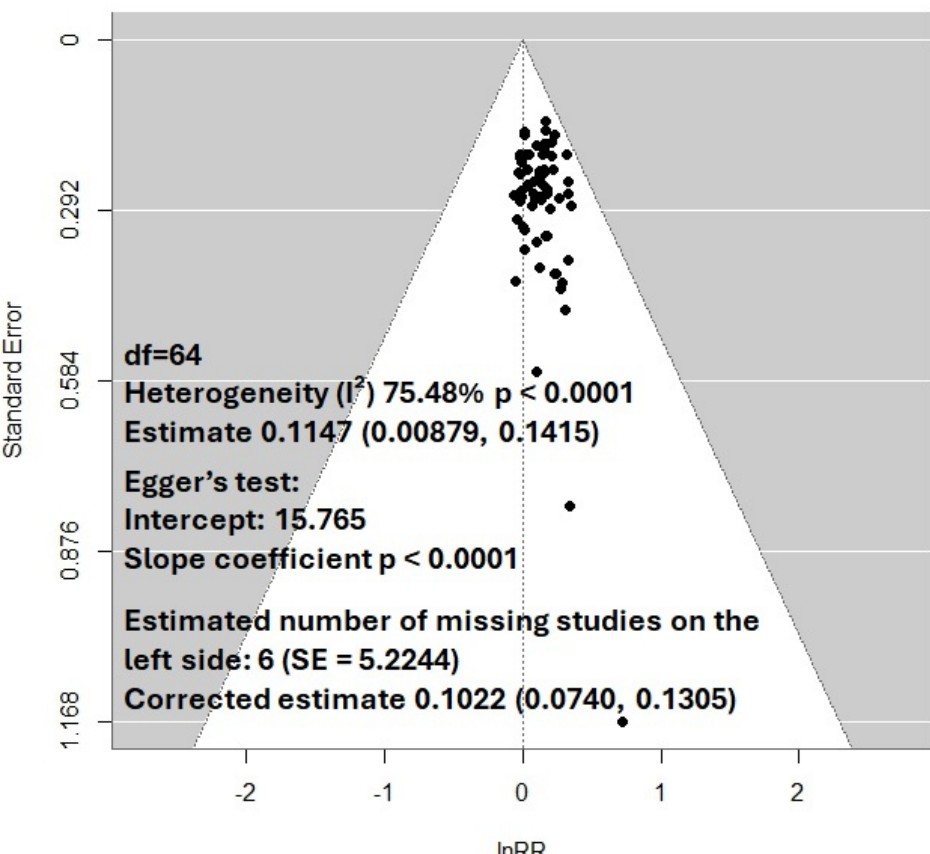

**Figure 2. Funnel plot after Xu et al. (2024).** Funnel plots are evaluated for symmetry: in the absence
of bias, they should resemble an inverted funnel, with larger (more precise) studies at the top and
smaller (less precise) studies scattered at the base. Asymmetry may suggest publication bias, such as
an overrepresentation of small studies with large effects due to selective publication of positive
findings. The triangle represents the 95% confidence interval, and studies outside this interval may
indicate heterogeneity ($I^2$) or bias. Heterogeneity reflects inconsistent results caused by variations in
study design, populations, interventions, or actual outcomes.

**iii) Methodological mismatch? Limited scalability of soil incubations and the need to differentiate priming effects from rhizosphere priming effects**

'Priming effects (PE)' refer to interactions between soils, soil microbes and added substances, while 'rhizosphere priming effects (RPE)' more specifically describe the interactions between living plant roots, their exudation and other rhizodeposition, rhizosphere microbes and rhizosphere soils. It is

important to distinguish between the two, because they differ in their driving factors and the scale of inference (Figure 3). Priming effects are caused by a static, sometimes repeated, source of substrate input, and usually measured in soil incubation. Rhizosphere priming effects describe changes in SOM mineralisation in the root zone, and are hence subject to dynamic changes in C and nutrient supply and demand, where the plant acts simultaneously as a sink for nutrients and water and a source of C.

Hence, several plant physiological parameters like rate of photosynthesis and root exudation are also determinant for rhizosphere priming effects (Dijkstra et al. 2013; Yin et al. 2018; Tang et al. 2019). It is important to acknowledge the limitations in the scalability of isolated soil incubations to ecosystem processes given that C, nutrient and water pools and fluxes are different in the rhizosphere of living plants as compared to reductionist lab incubations. Moreover, soil incubations are usually conducted

under standardised conditions of temperature and soil moisture, and usually soils are sieved before the incubation. Therefore, we have limited knowledge of priming effects in intact soils under variable environmental conditions, and cannot conclude about an impact of priming effects at ecosystem scale based on this data, esp. as the magnitude of priming is usually higher in soil incubations than in the field (Chen et al., 2023). Hence, it is crucial for future studies to assess whether estimates of priming

effect (PE) and mechanistic insights derived from soil incubations accurately reflect processes of rhizosphere priming effects (RPE) in natural ecosystems. Further, more efforts need to be made to measure priming in the field to have a better signal to noise ratio in in-situ studies.

**Conclusion**

While the soil C priming effect may have limited impact on the C cycle, it is still valuable to evaluate

the role of priming in ecosystem functioning, such as how it influences microbial substrate preferences, plant-driven substrate switches, and nutrient dynamics. We suggest a shift from a climate-only perspective to understanding the factors controlling positive and negative priming and their temporal shifts for enhancing plant-soil system resilience and overall ecosystem health. Priming papers should as a rule evaluate the net C balance by juxtapositioning the quantities of primed C and

added C to understand whether C has been lost from the system or not. Because often there is no net C loss from soil despite positive priming being reported. To reliably determine the direction of priming across several studies (meta-analysis), publication bias needs to be evaluated very carefully. And prior to that, publication of negative or no priming effects needs to be encouraged. Future studies should also investigate potential discrepancies between soil incubations and field experiments and could

address the potential to leverage rhizosphere priming effects to optimise plant nutrition. To upscale (rhizosphere) priming effects to ecosystem processes, their dependency on nutrient, water and temperature dynamics needs to be investigated, which is the opposite of laboratory soil incubations under standardized conditions.

**Figure 3. Critical checklist to contextualise study design.** Red circles indicate common approaches in most experiments. The intermediate paths risk to contain either too much ecological noise to obtain a mechanistic signal, or assume too many simplifications which trigger mechanisms which are rarely to occur in natural terrestrial ecosystems.

| SCALE OF INFERENCE / TERMINOLOGY | |
|---|---|
| Does the study involve a living plant? | |
| YES: Rhizosphere priming effect (RPE) | NO: Priming effect (PE) |
| Calculated direction of priming can change depending on whether a planted or unplanted control is used (Jian & Bengtson, 2022). Seasonality of plant growth can lead to fluctuating RPE (direction & magnitude), therefore high temporal resolution of measurements is needed (e.g. Diao et al., 2022; Schiedung et al., 2023). Depending on type and intensity of isotopic labelling (continuous or pulse $^{13/14}$C, $C_3C_4$-conversion), RPE estimates can carry uncertainty >100% (e.g. Cros et al. 2019). | Model to quantify SOM-dynamics under litter inputs or agricultural residual incorporation in absence of living plants. Single or repeated inputs of more or less diverse C/nutrient rich compounds are weak representatives of root exudates, which vary as a function of plant nutrient and water uptake and environmental conditions. Limited interpretability at ecosystem level as reductionist approaches struggle to represent realistic water and nutrient flows normally directed towards the plant (e.g. Raza et al., 2025). |
| Is the soil sieved (how many mm?), Are soil moisture and temperature kept within a given range (which range)? | |
| YES: Standardized, controlled conditions | NO: Natural conditions |
| Sieving changes soil fractions and baseline $CO_2$-emissions, may release C and nutrients, may break fungal hyphae, changes water dynamics (e.g. Datta et al., 2014; Even et al., 2025). | As RPE fluctuates with environmental conditions (and plant growth), high temporal and spatial resolution of RPE measurements may be required (e.g. Ma et al., 2012; Diao et al., 2022). |
| Is temporal variability taken into account? Over which timescale is soil mineralisation monitored? (How) is cumulative priming estimated? | |
| YES | NO |
| Risky to upscale RPE from snap-shot measurements; to identify required measurement frequency, future studies could monitor diurnal variation of RPE and/or variation in response to sun light/plant photosynthesis. | Limitations to the interpretability at ecosystem level arise as temperature and soil moisture in natural environments change on diurnal and seasonal scales. |
| Is spatial variability taken into account? | |
| YES | NO |
| To identify required measurement distribution, future studies could monitor spatial variation of RPE within and across given landscapes. | Limitations to the interpretability at ecosystem level arise as soil processes in natural environments can change on micro and macroscales. |

| CARBON BALANCE | |
|---|---|
| Is the amount of added substrate/plant-C inputs measured and reported? Is the amount of not-respired added substrate/plant-C inputs calculated and reported? Is the fate of not-respired added C known (biomass, DOC, plant re-uptake...)? | |
| YES | NO |
| Plant root C inputs to soil and their fate in soil are difficult to quantify / a knowledge gap, addressing this hence a lever to improve estimates of RPE (e.g. Pausch & Kuzyakov, 2018). Complementary measurements include plant photosynthesis and above and belowground plant biomass production. Dark $CO_2$-fluxes should also be taken into consideration. | Difficult to estimate in systems involving living plants, so the ability to calculate a net C balance is a strength of reductionist soil incubations. Should be facultative to report quantities of added-but-not-respired-C in addition to any priming effects, otherwise conclusions about net system C-loss or gain are not possible. |
| Is microbial biomass quantified (how often, in all modalities, incl. isotopic composition...)? | |
| YES | NO |
| Diverting opinions about how variable microbial biomass is, high temporal & spatial resolution may be needed. Alternatively, if the sum of inputs and outputs is known, net C balance can be calculated without resolving for the fate of C-inputs in different pools. | If the sum of inputs and outputs is known, net C balance can be calculated without resolving for the fate of C-inputs in different pools. Recycling of microbial biomass can lead to "apparent priming" (Blagodatskaya & Kuzyakov, 2008). |
| Is the emitted $CO_2$ separated into plant/substrate-source and soil-source? | |
| YES | NO |
| The is inevitable to calculate priming. For plant studies, uncertainty estimates need to be provided taking variability of molecular and isotopic composition of root inputs to soil into account (e.g. Ma et al., 2012) | If only $CO_2$ of soil-origin is reported, apparent priming cannot be estimated. Total $CO_2$ (soil and substrate derived) needs to be known to calculate a C balance of net inputs vs net outputs. |

**Data availability:** The data presented here is available in the cited papers and respective supplementary materials.

**Author contribution**: JM analysed the data and wrote the first draft. All authors critically evaluated the manuscript and approved the final version.

**Competing interests:** The authors have no conflicts to declare.

**Financial support:** The authors received no financial support for this study.

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
