# Peer review of "What if publication bias is the rule and net carbon loss from priming the exception?"

_EGUsphere, 2025_

## Author Response (AR1)

**egusphere-2025-1067 response to reviewers**

**Editor statement:**

Thank you very much for choosing SOIL for this interesting manuscript. I agree with the comments of the reviewers and in general with the response of the authors. Taking all the reviewer's comments and your responses into account, I would recommend major revisions of the manuscripts.

I particularly agree that the "the ideas developed must be more attractive and be inspiring for the community to change their ways of doing priming research." As reviewer 1, I have some doubts if re-analysis of the data from Xu et al. should be the main purpose of a forum article.

You wrote "We have some suggestions to improve the current methods, but the best ideas could probably come from an open debate, which is the primary goal of this forum article." I think that adding some ideas could be an excellent base for stimulating an open debate.

I strongly agree with your statement: The publication of negative and no priming effects needs to be encouraged strongly because some researchers do not publish negative priming because either they think it's a "wrong result" or because they think it has no value.

Please use all the very constructive comments of the reviewers for revising your article.

>> Thank you. To address reviewer comments, we made the following main changes:

- We include some graphics which hopefully inspire readers to reflect on the methods used and their won thinking process. In Figure 3 we also include some concrete steps which could help to make measurements R.PE more accurate (e.g. quantification of root inputs and their isotopic composition, quantification of diurnal & seasonal variation in rhizosphere SOM-mineralisation). We also include knowledge gaps and sources of uncertainty in priming estimates.
- We completely rewrote section ii) to reduce discussion of Xu et al. and discuss more generally how confirmation and publication biases can affect perception of priming effects, which can affect hypothesis formulation and data interpretation. This also addresses request by reviewer1 to provide more details on the effects of cognitive and systemic biases and propose some ideas how to avoid them.

In the revised version we maintained the word limit of the forum format (<2500 words), but we exceed the number of display items which should be n=3 but is n=4. Depending on editor & reviewer advice we are happy to re-adjust this. As we went deeper into the discussion, we also extended the reference list accordingly.

In the tracked-changes document we also highlight all changes in yellow.

**Reply to RC1: 'Comment on egusphere-2025-1067', Anonymous Referee #1**

The paper submitted by Michel et al. explores priming effects in soils, which refer to the influence of labile carbon inputs on the mineralization of organic matter by microbes. These effects can be positive (increased mineralization) or negative (reduced mineralization). The article highlights three key aspects to better understand these effects in their ecological context: (i) Evaluation of net carbon balances (ii) Publication bias and (iii) Difference between general priming effects and rhizosphere-specific priming. The article calls for a more nuanced approach to priming research, encouraging the publication of studies on negative or neutral

effects, carefully evaluating publication biases, and distinguishing general priming effects from rhizosphere-specific effects. It also emphasizes the importance of conducting field studies to better understand these phenomena under natural conditions.

I think the points developed by the authors are fair and worth to be published but the current version of the paper is a bit disappointing since it skims over the issues without going into depth. It may come from the format which is not very clear. Is it a review or is it an opinion paper? If this is a review, much more literature must be cited if this an opinion paper, the ideas developed must be more attractive and be inspiring for the community to change their ways of doing priming research.

>> We thank the reviewer for their time and feedback on this manuscript. We apologies if the format is confusing, it is a "forum article" and we adhere to respective journal guidelines (https://www.soil-journal.net/about/manuscript_types.html). We revised the manuscript to include ideas how to improve priming studies and propose some items every priming paper should report to reduce the risk of misinterpretations (Figure 3).

For instance, the authors wrote that "there is little empirical evidence for net C losses", I fully agree with the statement but it should be better explained why priming is sometimes see as a mechanism leading to net losses and show some papers that suggest it and explain why they are mistaken. So far the authors mostly cite papers that do not observe any net loss.

>> We clarify this point in the revised manuscript (lines 61-66). We can add references here of papers reporting positive priming & C loss, but the average reader will likely already be familiar with those studies. We introduce positive priming in line 33, with the same attention as negative priming, but the idea is really to introduce the reader to the more hidden side of priming and the lack of C balances.

The second section is simply a re-analysis of the data from Xu et al, which merely confirms the conclusions of the original paper. In my opinion, this does not add much to the message of Xu et al. More details on the effects of bias should be provided and some ideas to avoid them should be also proposed.

>> In our opinion an average estimate of 10.7% PE (estimated effect size (log-transformed response ratio) of 0.1022 (CI95: 0.0740, 0.1305) is too close to zero priming to state a global rule about the direction of priming, especially given that none of the underlying meta-analysis has been corrected for publication bias. Our re-analysis dropped the author's priming estimate of +37% to +10.7%, which is 2/3 less than the original estimate (an over-estimation of 60%). It is a huge difference. The fact that they reference a +PE of 125% despite knowing about publication biases etc. is just a nice example how the authors themselves are victim of a number of biases we now mention in Table 1.

Anyway, we completely rewrote section ii) and significantly cut down the re-analysis of Xu et al. (mainly in Figure 2 now).

Then the third section is mainly focused on the importance of using intact soils instead of sieved soils whereas the title of the section suggests that RPE and PE should be treated differently. So far this section is not really convincing mostly because it lacks more clear examples and it needs some suggestions to improve the current methods.

>> Thanks, we added examples and food for thought to this section by means of Figure 3.

**Reply to RC2: 'Comment on egusphere-2025-1067', Anonymous Referee #2**

In their manuscript with a promising title "What if publication bias is the rule and net carbon loss from priming the exception?" the authors aim to address an important aspect of the relevance of priming. The main point is that priming (i.e. enhanced or reduced mineralization of native soil organic carbon) occurs as an elemental process but the net carbon balance is in most cases not considers in the literature. I agree that this causes a biased understanding of the priming in relation to total carbon changes of soils. In fact, I agree that priming, especially positive, is not on relevant magnitudes to off-set the actual input that is required to initiate priming in the first place. I also generally agree that our understanding is biased due to studies that used "artificial" organic matter inputs and that we have a limited understanding of the complex effects of the whole rhizosphere effect and associated organic matter inputs but also biogeochemical shifts. Overall, I agree that such statements need to be made clearly in order to facilitate future research focuses. However, I have to be honest that I was a little disappointed that the authors remain on a very broad level with their discussion. In addition, it is not clear if this is supposed to be a review or opinion and I miss any clear statement how we should move on.

>> We thank the reviewer for their time and feedback on this manuscript. We apologise for the brevity and confusion about the "forum" format and are more specific about possible avenues of future research in Figure 3 (which requires that we critically evaluate the conclusions of past research which is neglecting negative priming).

The first chapter summarizes very briefly that positive priming is not affecting the net carbon balance.

The second chapter re-analysis the study presented by Qin et al (2024) and Xu et al. (2024). In fact, the authors repeat the statement by Qin et al (2024) who stated that the net balance is mainly positive. Therefor, this is supporting the authors point but not providing new information that is shown in Figure 1a that only shows the included studies in Qin et al. Regarding the Xu et al. (2024) study, the chapter in the manuscript is quite technical how the authors re assessed the data and what funnel plots are (line 93-123, which is a lot regarding the short manuscript). The authors provide an improved interpretation of the same plot that is discussed by Xu et al in the supplement figure 2. The authors apply the standard error on ln transformed the priming effect observed in the studies. Technically this seems correct. However, this seems to be rather a response to the Xu et al paper than making a new point. In fact, the Xu et al in the supplement figure 2 already shows the asymmetry that most studies are reporting positive priming effects, when I visually assess the deviation from x=0 in the plot (see plot below). The authors finally estimate that only six studies with moderate or negative priming effects are need to overcome the publication bias. They state in line 137 "It would be interesting to recalculate a global PE estimate from the primary research data of all underlying meta-analysis corrected for publication bias.". This raises the question why the authors are not providing this. Finally, the authors just encourage to publish negative and moderate priming effects.

>> The point we tried to bring forward is that the paper by Xu et al. seems to be affected by all the biases we mention in our new Table 1, as they clearly set out to "prove a rule" (that of positive priming), while their data actually only provides weak evidence for that with an estimate of 10.7% PE (estimated effect size (log-transformed response ratio) of 0.1022 (CI95: 0.0740, 0.1305) which is based on meta-analysis that are not corrected for publication bias; in fact some of the meta-analysis explicitly excluded negative PE values. With hypothesis specifically

embracing negative priming, different conclusions could have been drawn from the same data, but probably less catchy and not in line with the dominant narrative. Further, for ecological context it is imperative to report a net C balance which the authors failed to do. All that the paper does is re-emphasise the dominant view (e.g. by stating a +125%PE in the abstract for example which they know is an outlier and not representative for the overall picture). Anyway, we completely rewrote section ii) to focus on the common biases and how they may affect researchers who study priming effects.

The third chapter claims that rhizosphere priming and general priming are different. Yes, it is important that we include more complexity and understand the rhizosphere effects better. It is a major challenge to study such effects in intact systems that include the whole rhizosphere. This is already discussed in other studies and should be clear. The authors just state that we need to understand rhizosphere priming in natural ecosystems, but they do not provide any way forward to do so. Therefore, I do not see any substantial addition to the discussion based on the very broad and short chapter here.

>> The reason why we mention this is that the magnitude of priming in soil incubations is higher than in experiments involving living plants. If we jump form soil incubations to ecosystem processes, we risk overestimating priming. But to date, most of our priming knowledge is based on soil incubations. For example, from the 12 meta-analyses compiled in Xu et al. (2024), n=9 report data exclusively from isolated soil incubations, two included meta-analysis comprise a mix of lab and pot/field experiments (Wang et al., 2016; Feng & Zhu, 2021) and one meta-analysis summarized exclusively experiments with living plants (Huo et al., 2017).

We hope that people reading this forum article will become curious about what priming is like when they actually include a plant. We also added figure 3 to invite readers to critically evaluate the common methods and limitations. We highlight knowledge gaps and data which could help improve estimates of priming at ecosystem scale (e.g. addressing diurnal variability and isotopic fractionation in planta which results in variable isotopic signature in root exudates, propagating into large uncertainty of RPE estimates).

We are aware that most readers in SOIL are probably proponents of soil incubation studies and don't like to hear these are flawed and they should do more complicated experiments, but we dare to highlight that for a study of rhizosphere priming effects, a rhizosphere has to be present in the experimental set-up.

In addition, the authors do not provide any clear definition of the priming effect in the first place. For example, in line 15 it is stated: "Priming effects in soil science describe the influence of labile carbon inputs on rates of microbial soil organic matter mineralisation, which can either increase (positive priming) or decrease (negative priming)." Therefore, they miss the important aspect that priming is affecting the native or present soil organic matter after the addition of new organic matter.

>> Thanks, we clarified this point in the revised manuscript (line 16)

---

## Author Response (AR2)

Responses to suggestions for revision 2 for EGUSPHERE-2025-1067 | Forum article

The authors provided a well revised version of their manuscript "What if publication bias is the rule and net carbon loss from priming the exception". They replied to all my raised concerns and points and the current version is well suitable as a forum contribution. It is timely to stimulate the discussion and future research regarding priming effects. The new figure and table help to convey the authors aspects.

I make some more specific comments below.

Beside these comments, I wonder if the authors could be clearer regarding the final conclusion they make in the abstract. This is coming short in the manuscript but to me a relevant point. I would even support a more provocative statement. As the authors discuss, priming is conceptually studied a lot as a C and climate effect. This was over the past few decades the main focus. At the same time, we have the bias that it is mainly positive and the perception was built that priming is always an emission source of soils. I would argue that in real ecosystems and not experimental set ups that focus on C fluxes only, priming is an essential mechanism for ecosystem function. Thus, I would argue that priming studies focusing on the more holistically on the full effect in soils and not on the less relevant C effect only. This is in the manuscript, but could be clearer. I specified this below also for the conclusion.

>> We revised the final conclusion in the abstract to be clearer that priming is not only about C-cycling, but about ecosystem functioning (lines 27ff).

This does not mean that C addition experiments are not useful at all. They are a tool to understand the soils responses and relate different nutrient and energy effects between soils at similar conditions. It is still a useful tool. A tool that should start to be more complex by adding full rhizosphere effects.

Regarding the display items, I would suggest to allow the four items. However, if this is not possible, I would consider to move the Figure 2 (Funnel figure) to the SI. In addition, this figure might require some improvements in the visualization. I appreciate Fig. 3 as this supports the authors points very well.

>> Thank you, we're happy to keep the four display items.

Title: I think the grammar and readability should be improved. I suggest "What if publication bias is the rule, and priming is the exception". Thus. Adding a "," before the "and" and a second "is" before "the exception".

>> We'd like to keep the longer version, because it is not priming which is the exception, but carbon loss from priming.

Specific comments:

Line 15 and throughout the manuscript: "Labile" is not well defined and can be confused with some specific pool/fractions. I would recommend to use "fast cycling", which is not more precise in terms of dynamics but better in terms of the meaning regarding its mineralisation. In fact, right here, it would be possible to just remove labile.

>> Here and elsewhere, we changed "labile" to "fresh" or removed completely.

Line 22 and whole manuscript: Make sure to be consistent in the use of element names (ie. carbon) or symbols (ie. C)

>> Checked and corrected throughout. We write "carbon (C)" on first mention in the abstract and main text. We keep the full word "carbon" in headings and in table 1 we also keep the full word if it is in direct speech. Otherwise, we abbreviate consistently.

Line 28-29: "nutrient cycling and plant nutrition" could be rephrased to "ecosystem and soil functioning" or similar to align with more the broadscale effect.

>> We rewrote this sentence to read "Future research should focus on scalable experiments linking priming to plant nutrition via C, nutrient and water cycling to understand priming in context of ecosystem functioning." (lines 27ff).

Line 32: I would also argue that not only root exudates but also root litter is important here. I would make clear that root exudates are the active pathways for roots to initiate or inhibit priming.

>> We rewrote these sentences to be clear about the sources of the main C-inputs: root exudates, root litter, leaf litter (lines 31ff).

Line 23: See comment regarding "labile", I would suggest here "...relative fast cycling carbon compounds compared to existing SOC can .... "

>> We rephrased this sentence to omit "labile".

Line 41-44: It is missing here that rhizosphere input can also be directly in interaction with the mineral surface (e.g Sokol et al. 2019, 2024). This depends on biotic as well as abiotic conditions. The authors could add here the degree of complexity, that we do not fully understand the abiotic effects as well and this will ultimately affect the priming responses, depending on soil conditions.

Sokol, N.W., Foley, M.M., Blazewicz, S.J., Bhattacharyya, A., DiDonato, N., Estera-Molina, K., Firestone, M., Greenlon, A., Hungate, B.A., Kimbrel, J., Liquet, J., Lafler, M., Marple, M., Nico, P.S., Paša-Tolić, L., Slessarev, E., Pett-Ridge, J., 2024. The path from root input to mineral-associated soil carbon is dictated by habitat-specific microbial traits and soil moisture. Soil Biology and Biochemistry 193, 109367. https://doi.org/10.1016/j.soilbio.2024.109367

Sokol, N.W., Sanderman, J., Bradford, M.A., 2019. Pathways of mineral-associated soil organic matter formation: Integrating the role of plant carbon source, chemistry, and point of entry. Global Change Biology 25, 12–24. https://doi.org/10.1111/gcb.14482

>> Thank you, we added this aspect (line 49-51).

Line 43: with respect to my comment of the use "labile", "complex" refers here to rather slow cycling organic matter.

>> Here and elsewhere, we changed "labile" to "fresh" or removed completely.

Line 88-90: It might be useful to extent a little bit here and say what the misleading means. Basically it resulted in the perception that positive priming results in soils as emission source.

>> Yes, we rephrased this to be clear that priming if often misleadingly presented as net C loss (now line 91ff).

Table 1: is a great addition and valid for the discussion and forum format. The Hindsight bias example should be more focused on priming to align with the other examples. Like: "In knew that positive priming will occur"

>> Thank you. We would like to keep "I knew that would happen" as example here, because it is such a common and relatable phrase that almost everyone will find themselves having said that at some point in their lives. Therefore, we think this sentence will help the readers to truly self-reflect. No-one is immune to biases, so it is important to raise bias awareness, and we think this generic examples could help in this regard.

Line 103: Link to Fig. 2 is not clear here as the funnel plot is discussed later, where the link is missing.

>> We moved the reference to figure 2 later in the text where it ties directly to publication bias and funnel plots (line 129) and clarified the funnel plot is of a meta-meta analysis and does not correct underlying meta-analysis (lines 139/140).

Line 109-113: What specific studies to the authors mean here?

>> It may vary from reader to reader which study in particular got caught in their mind as the "top of mind" reference, so we don't want to point to any particular paper here. Similarly, there are numerous "common studies" reporting varied priming effects, so we don't necessarily want to highlight any individual studies here. These lines aim more to introduce the reader to the concept of the availability heuristic, where one study gets "hyped" to represent a situation, while the majority of the body of literature, which might point to more nuanced circumstances, gets ignored.

Line 176-178: Also more efforts to measure in-situ priming. My personal experience is that experiments in fields fail (I had two studies with isotopic material and not even growing plants, not publishes with any priming focus in the end) because the even very small-scale variability in soils (e.g. cores in <20cm vicinity) makes it extremely difficult to separate relatively small changes by priming from SOC variability. Not to mention any shifts in conditions in treatments compared to control (e.g. all roots grow in the added material to mine nutrients and no shift in controls). This means a change in input of organic matter that is not easy to account for. We need to manage to have a better signal to noise ratio in in-situ studies. I would argue that this is more important than linking incubation studies to real ecosystems.

>> We added this aspect (lines 181/182).

Conclusion:

Personally, I think the question "do we need to care at all about priming?" is interesting and could be asked here.

In terms of C changes it might not be interesting, as proposed here. Thus, one could say we do not need to care at all. However, and maybe more importantly, priming has an important role in ecosystem functioning. The role of this for resilience and overall functioning of the soil-plant system is valuable to understand. Meaning, what substrate do microbial communities prefer, how doe plant affect the substrate-switch of microbes and for which benefit (nutrients and not C).

Therefore, I would argue to shift form a climate only perspective on priming to an ecosystem function of priming and which factors control positive and negative priming as well as temporal shifts of positive and negative priming.

>> We extended on this point in lines 184-188.